# Dual-Task Performance, Balance and Aerobic Capacity as Predictors of Falls in Older Adults with Cardiovascular Disease: A Comparative Study

**DOI:** 10.3390/bs13060488

**Published:** 2023-06-09

**Authors:** Heitor Silveira, Juliana Lima, Jessica Plácido, José Vinícius Ferreira, Renan Ferreira, Jerson Laks, Andrea Deslandes

**Affiliations:** 1Instituto de Psiquiatria, Universidade Federal do Rio de Janeiro, Rio de Janeiro 22290-140, Brazil; hdssilveira@gmail.com (H.S.); julianadias94@gmail.com (J.L.); jessica.placido10@hotmail.com (J.P.); joseviniciusalvesferreira@gmail.com (J.V.F.); jersonlaks@gmail.com (J.L.); 2Instituto Nacional de Tecnologia, Rio de Janeiro 20081-312, Brazil; baltarrenan@gmail.com; 3Clínica da Gávea, Rio de Janeiro 22451-262, Brazil

**Keywords:** older adults, cardiovascular disease, physical function, dual-task, polypharmacy, executive function

## Abstract

Cardiovascular diseases (CVD) are highly prevalent and strongly associated with the risk of falls in the elderly. Falls are associated with impairments in cognition and functional or gait performance; however, little is known about these associations in the elderly population with CVD. In this study, we aimed to clarify the possible associations of physical capacity and functional and cognitive outcomes with the incidence of falls in older adults with CVD. In this comparative study, 72 elderly patients were divided into fallers (*n* = 24 cases) and non-fallers (*n* = 48 controls) according to the occurrence of falls within one year. Machine learning techniques were adopted to formulate a classification model and identify the most important variables associated with the risk of falls. Participants with the worst cardiac health classification, older age, the worst cognitive and functional performance, balance and aerobic capacity were prevalent in the case group. The variables of most importance for the machine learning model were VO_2max_, dual-task in seconds and the Berg Scale. There was a significant association between cognitive-motor performance and the incidence of falls. Dual-task performance, balance, and aerobic capacity levels were associated with an increased risk of falls, in older adults with CVD, during a year of observation.

## 1. Introduction

Globally, the proportion of the population aged 65 years or over increased from 6% in 1990 to 9% in 2019, with a projected rise to 16% by 2050, at which point one in six people in the world will be aged ≥65 years. Furthermore, projections indicate that nearly 80% of older people will be living in low and middle-income countries in 2050 [1]. These data highlight the importance of discussing healthy aging by developing and maintaining the functional abilities and intrinsic capacity that enable well-being in older age [2,3]. These functional abilities are impacted by health conditions that are common in older adults including musculoskeletal and sensory impairment, cardiovascular diseases (CVDs), hypertension, mental disorders, cognitive decline, and geriatric syndromes such as frailty, which can increase the risk of falls and disability in this population [2].

Considering the prevalence of these conditions, falls are one of the most common causes of injury in people of all ages, and also an important risk for mortality, especially in the elderly [4]. Falls were noted to be the second leading cause of unintentional injury-related deaths worldwide [5], and represent a major public health burden given the high incidence of hospitalizations [4,6]. Each year, an estimated 684.000 worldwide die from falls, with the majority of fatalities occurring in older adults [5]. Of the total global occurrence of deaths from falls, more than 80% occur in low and middle-income countries [2]. Due to the high cost to the health system, identification of the risk factors associated with the incidence of falls is crucial to provide practicable guidelines for the prevention of episodes and management of care [7].

Since falls are a complex event with multiple causes, the increased propensity for falling in older adults results from the interaction of risk factors. These factors can be categorized as modifiable or non-modifiable and divided into four dimensions: biological, socioeconomic, environmental and behavioral. Among the non-modifiable factors, age, sex and the presence of genetic biomarkers are biological factors involved in the increased risk of developing cognitive/physical decline. The modifiable risk factors include behavioral influences such as multiple medication use, alcohol intake, and lack of exercise [8]. Both the biological and behavioral risk factors are influenced by some components of health that are modulated by physical exercise, which improve balance, functional capacity, lower limb strength, intrinsic capacity, and prevent chronic illnesses, thereby preventing falls [9].

In older adults with CVDs, falls may have multiple causes such as sensorial, cognitive and physical impairments [10,11]. These health conditions are also associated with the use of multiple medications that can also increase the risk of falls since they may cause nausea, dizziness, muscle weakness, or blurred vision [12]. The drugs that have most commonly been associated with falls are diuretics (odds ratio (OR)) = 1.08; 1.02–1.16), digoxin (OR = 1.22; 1.05–1.42), and type IA anti-arrhythmic medications (OR = 1.59; 1.02–2.48) [13]. Furthermore, Denfeld et al. (2022) showed that 61% of patients with CVDs had a high risk of falling, indicating that awareness of this issue is essential to provide prevention strategies in hospital settings [12]. The physical impairments caused by CVDs, such as abnormal gait and balance, are another important cause of falls in older adults with these conditions. Studies have also highlighted that both balance and gait impairments can be influenced by polypharmacy, creating two paths for the possible modulation of physical capacity in the risk of falls in this population. However, few prospective studies have focused primarily on falls among people with CVD, with most data obtained from secondary analyses or supplementary injury medical records [12].

Recently, the American Heart Association (AHA) published a statement on the prevention and management of falls in people with CVD, providing a list of suggested targets and assessment criteria including frailty, gait, standing balance, proprioception, and cognitive function [12]. A decade earlier, the National Institute for Health and Care Excellence (NICE) guidelines for prevention of falls in older people focused on monitoring balance and gait deficits [14]; however, neither the NICE nor the AHA statements highlighted dual-task (DT) assessments as a possible target for fall prevention. DT assessments are defined as the ability to perform two tasks simultaneously, and these tasks may be constituted by two types of motor (motor-motor tasks) or cognitive (cognitive-cognitive tasks) demands or even a combination of these two stimuli. The DT assessment represents a paradigm of daily life and is often used for investigating the cognitive processing of motor behavior. DT impairment is associated with the prevalence of cognitive decline and dementia, specifically Alzheimer’s Disease [15,16]. DT capacity is also affected by the presence of stroke [17] and hypertension [18]. In addition, interventions based on DT training in older adults with a history of stroke have been effective in reducing the risk of falls and fall injuries [19]. Pang et al. (2018) reported that a 60 min session of DT, balance, and mobility training each week for eight consecutive weeks reduced the risk of falls by 25.0% and the risk of fall-related injuries by 22.2% in older patients with stroke. However, to our knowledge, the specific relationship between DT and fall risks in older adults with other types of CVD has not yet been investigated [20].

Fall-related mechanisms involve executive control and some motor functions, such as balance and gait [21]. Several studies have identified an association between performance in executive function tasks, especially response inhibition and selective attention skills, and the risk of falls [21,22,23]. This association may be mediated by lesions in the white matter of the frontal areas of the brain, with injuries affecting sensory and motor functions that increase the risk of falls [24]. In their theory of the central benefit model, Liu–Ambrose et al. (2012) postulated that the improvement of cognitive function, specifically executive functions, is associated with functional plasticity, which is the main mechanism by which exercise reduces falls in older adults. Decreased executive functions are associated with a higher risk of falls because of the balance and gait impairments caused by the attentional deficits and also by diminished self-regulation, motivation and initiation. Conversely, the loss of motivation and initiation negatively impacts balance and gait, which can lead to reductions in executive functions via a feedback loop [21]. This proposal also raises the hypothesis that variables, such as cardiovascular capacity, may play a role in preventing falls through their effects on cognitive and vascular reserve [25].

In recent decades, studies have revealed the critical importance of attention and executive function in motor tasks, especially in walking, thus superseding the view that gait is automatic and occurs without cognitive demand. The demonstration by Lundin-Olsson et al. (1997) that frail older adults stop walking when they need to talk, and that this mechanism is associated with an increased risk of falls over a six-month period, highlighted the importance of evaluating the role of DT ability (gait performance declines during the simultaneous execution of a secondary cognitive task) in the process of falls [26]. Currently, the literature points to an association between worse DT completion and an increased risk of falls in both healthy older adults and those with cognitive decline, but little is known about this relationship in older adults with CVDs.

Given that many older adults have a type of CVD and take multiple medicines for clinical treatments, investigation of possible modifiable factors is crucial to prevent the incidence of falls in elderly individuals with cardiovascular conditions. Machine learning techniques have proved invaluable for investigating preventive factors/markers related to health. Studies have shown the potential of these methods for improving clinical diagnoses and developing strategies to prevent and treat diseases during the course of aging. In this study, we adopted machine learning techniques to investigate the possible associations of physical capacity and function and cognitive outcomes with the incidence of falls in older adults undergoing cardiac rehabilitation treatment.

## 2. Materials and Methods

### 2.1. Study Design, Setting and Participants

In this comparative study based on a nested case-control design, older adults (>60 years; both sexes) with a clinical CVD diagnosis according to the International Disease Classification CID-10 [27], were recruited from two clinical fitness centers for cardiac rehabilitation and exercise training in Rio de Janeiro, Brazil.

The inclusion criteria were as follows: being classified in stages B or C of a cardiovascular condition according to the American College of Cardiology and the American Heart Association (ACC/AHA) standards [28,29], without recurrent symptoms. The following exclusion criteria were also applied: being classified in stages A (risk of CVD) or D (severe condition); having cardiac pacemakers; being illiterate; having neurological or mental disorders; having physical or vestibular impairments that inhibit a 10-m walking test; or having another severe comorbidity.

The case group (fallers) was classified by the incidence of falls in the sample investigated for 12 months. The control group (non-fallers) was composed of individuals that did not report the occurrence of falls in this period. The fall event was defined according to the World Health Organization guidelines (2021), which considers an occurrence of fall when a person comes to rest on the floor or other lower level unintentionally.

Generally, the patients attended the fitness center two or three times a week to undertake physical exercises supervised by a professional. The cardiovascular health assessment was performed using a structured clinical interview following ACC/AHA standards [28,29]. The VO_2max_ was estimated with a treadmill ramp protocol starting at a low speed and progressively increasing the exertion intensity though the angle of ramp incline [30]. All participants agreed to their involvement in this research and provided written informed consent. This study was approved by the Research Ethics Committee of the IPUB-UFRJ, under registration CAAE: 22532213.4.0000.5263, and by the Brazilian Registry of Clinical Trials—ReBEC, under U1111-1159-9394.

### 2.2. Procedures

First, we analyzed the patients’ clinical records for details of cardiovascular health classifications and VO_2max_ measurements; both assessments were conducted by a specialized physician. If patients were classified in stages B or C of cardiovascular conditions and had clinical authorization to exercise, we invited them to participate in this study. After first contact, they were scheduled for evaluation at the fitness centers where they were already monitored. All the participants signed the informed consent form before the evaluations began, and there was always an on-call physician at the call center for emergency support. All the tests were conducted by physical exercise professionals. The battery tests were performed in three parts and lasted approximately 1 h.

Before the assessments began, we measured resting blood pressure (BP) and heart rate (HR), using the international protocol [31] with a Littmann cardiology stethoscope (Littmann^®^, Saint Paul, MN, USA), a sphygmomanometer (Welch-Allyn^®^ Tycos, New York, NY, USA), and a finger oximeter (AccuMed^®^, Rio de Janeiro, RJ, Brazil).

Participants were then interviewed to assess sociodemographic data, physical activity level, disease duration, routine medication use, fear of falling, and cognitive and aerobic fitness status. To assess the fear of falling, we used the Falls Efficacy Scale—International Questionnaire (FES-I) [32]. For cognitive status, we applied the Mini-Mental State Exam [33], the trail-making test, the verbal fluency test [34] and the Geriatric Depression Scale [35]. To verify the aerobic fitness status, we used the Veterans Specific Activity Questionnaire (VSAQ), which estimates the aerobic capacity expressed in metabolic equivalents (METs) [36]. The assessment of anthropometric factors (weight and height) was performed with a digital scale and a stadiometer PL 200 (Filizola^®^, São Paulo, SP, Brazil) and the body mass index (BMI) was calculated.

The participants also performed four functional tests from the Senior Fitness test battery (sit-and-stand test, 8-foot up-and-go (8UG) test, upper and lower limb flexibility tests) [37]. DT ability was evaluated through the combination of the 8UG and verbal fluency tests. Finally, the participants and their relatives were instructed to report, as soon as possible, the occurrence of falls during the next 12 months. In addition, during the weekly routine care in the fitness centers, the clinical staff monitored the participants’ occurrence of falls, and the researchers contacted them once a month to investigate the reported events.

### 2.3. Statistical Analysis

All statistical analyses were conducted in two steps: the descriptive analysis of demographic and clinical data, and the machine learning techniques. Kolmogorov–Smirnov and Levene’s tests were applied to verify the normality and homoscedasticity of the samples, respectively. To assess the differences between the groups, we used the *χ*^2^ test for categorical variables, the independent *t*-test for parametric variables, and the Mann–Whitney U-test for non-parametric variables.

For the machine learning tests, we first conducted an exploratory analysis to identify the presence of missing data in the data set. Next, the observations that had missing values were removed from the data set, resulting in a data frame of 52 observations and 40 variables. Before training the classification model (i.e., random forest), a subset of variables was selected from a combination of recursive feature elimination (RFE) and random forest (RFE-RF). This procedure was performed to reduce the dimension of the problem by removing variables that do not contribute enough to the proposed classification model. The choice of the best subset of features was based on the performance metric of the classification algorithm. Cohen’s kappa was the metric chosen to be optimized during this process, instead of the standard metric (i.e., accuracy) for classification problems. This was due to the imbalance of the data in the groups of interest (i.e., control and case).

Subsequently, the predictive analysis was conducted by applying an RF algorithm to classify the observations of the reduced data set. Cross-validation was performed during this process (*k* = 5) and two models were trained. The first model was developed based on Cohen’s kappa metric, while the second used the F-score metric in an attempt to reduce the number of false negatives (FNs) (i.e., cases classified as control), thus prioritizing the recall metric.

Since the most important variables were identified in the machine learning analyses, a logistic regression model was used to calculate the odds ratios (ORs) with 95% confidence intervals (CI) of this subset related to the risk of falling. To guide the selection of the possible adjustment variables, a causal diagram (DAG) was constructed using DAGitty software, version 3.0 (Appendix A). All statistical analyses were performed using R Studio^®^ version 3.6.0, and *p* ≤ 0.05 was considered to indicate statistical significance.

## 3. Results

### 3.1. A Comparative Design Based on a Nested Case-Control by Density Approach

Of the 210 older adults enrolled at the two fitness centers, 142 were contacted by telephone. Of these, 32 refused to participate in the study, and 38 were excluded because of physical mobility problems and major CVD; 72 participants were finally included in this study. In total, 24 falls were reported during the 12-month observation period, representing the case group; 48 control participants were selected (Figure 1). The case and control groups were matched by age, sex, and assessment site (centers).

### 3.2. Descriptive Analysis

The sample demographic and clinical characteristics are presented in Table 1. We had a high and similar prevalence of male and physically active older adults in both the case and control groups, with no significant difference between the groups. There were also no differences between groups in terms of the educational level, time of CVD duration, BMI score, depressive symptoms in the Geriatric Depression Scale, cognitive functions in the Mini-Mental State Exam, executive functions in the trail-making A test, words DT assessment, handgrip strength, Veterans Specific Activity Questionnaire results, and upper and lower limb flexibility tests.

There were significant differences between the groups in terms of the prevalence of subjects in the ACC/AHA classification category C, age, verbal fluency, trail-making B, DT, 8UG test, sit-and-stand test, Berg Scale, VO_2max_, and the total number of medicines taken (case: 6.5 [2.0–12.0]; control: 6.0 [2–17.0]; *p* = 0.05). These results showed that, compared to the control group, the case group had worse cardiac health as well as poorer cognitive and functional capacity, were older and, in general, took more medicines.

Regarding cardiovascular health, there was a significant difference between groups in terms of the prevalence of cardiac medication use (*p* < 0.001). Nevertheless, there were no statistical differences between the groups in terms of hypertension condition (case: 71%; control: 83%; *χ*^2^ = 1.51; *p* = 0.21) and the incidence of the following coronary diseases (*χ*^2^ = 81.46; *p* = 0.22) myocardial infarction (case: 17%; control: 10%), cardiac insufficiency (case: 17%; control: 31%), myocardial ischemia (case: 13%; control: 8%), aortic aneurysms (case: 4%; control: 4%), angina pectoris (case: 0%; control: 8%) and arrhythmias (case: 42%; control: 38%).

### 3.3. Machine Learning Analyses

In the machine learning analyses, the RFE-RF defined a subset of five variables as follows: Berg scale scores, type of cardiovascular medication used, DT in seconds, anticoagulant medicine taken, and VO_2max_. This subset was then used to train the classification model. The best model defined by cross-validation for the RFE-RF achieved values of 0.71 and 0.35 for accuracy and kappa, respectively. The sensitivity (i.e., correctly classified cases) of this model was 0.52.

Considering this result, a second model was trained to improve the classification of individuals in the positive class. For this purpose, the F-score metric was used in an attempt to improve the model’s performance. The default value of the F-score is 1 (i.e., precision and recall have equal priorities). On the other hand, an F-score > 1 means that prioritizing the correctness of the positive class will decrease the FN (i.e., the recall metric). Finally, the model trained based on an F-score of 1.5 achieved values of 0.76 and 0.50 for accuracy and kappa, respectively. Furthermore, the sensitivity of the model increased to 0.68.

In terms of overall values, the variables of most importance for this model were VO_2max_ (100.000), DT in seconds (94.561) and Berg Scale (93.909), in addition to the use of cardiac drugs, individually or combined, particularly β-blockers + anticoagulants + diuretics (36.426), anticoagulants only (35.987), β-blockers + antihypertensive (17.390), β-blockers + anticoagulants (7.764) and β-blockers only (0.000) (Figure 2).

### 3.4. Odds Ratio Analyses

Finally, we applied a logistic regression model to the three most important variables (overall values > 90.000) identified in the RFE-RF as VO_2max_, the DT in seconds, and the Berg Scale score (Table 2). The OR analyses showed a significant association between the Berg Scale and the risk of falls in older adults with CVD.

## 4. Discussion

In this study, we investigated the possible associations of physical and cognitive performance with the risk of falling in older adults with a clinical diagnosis of CVD. Although the case group showed worse semantic memory, executive function, and lower limb strength, the results of the machine learning analysis indicated that the occurrence of falls was mainly related to lower aerobic capacity levels, poorer balance and poorer DT capacity. Older adults with the lowest balance had a 69% higher risk of falling than their peers with a better performance in the Berg test. Moreover, we identified a significant difference between groups in the ACC/AHA classification and the total amount of medication used, with more individuals classified as having worse CVD and more medicines taken in the case group compared with the control group.

It is important to highlight the poorer physical function (balance, gait, and aerobic capacity) observed in the case group compared to that in the control group. Recently, Ye et al. (2020) used a machine learning approach to establish a model that predicts falls in older adults with 80% accuracy. The majority of the fall predictors in the model was composed of diagnoses of chronic diseases, especially the cardiovascular types, and physical dysfunctions, such as gait and mobility impairment [38]. Older adults with CVD had a 95% greater risk of falling, whereas the presence of abnormalities in gait and balance was associated with a four-fold increase in the risk of falling. The machine learning model established in our study also identified the importance of physical or mobility conditions in the occurrence of falls, and our comparative analyses also demonstrated a higher incidence of gait impairments in the case group. Moreover, aerobic capacity (measured by VO_2max_) was one of the most important variables that differentiated the case and control groups. Studies have demonstrated the relationship between VO_2max_ and lower limb strength, which are important for the maintenance of balance [39]. These results also confirm the importance of maintaining physical activity levels in order to prevent the occurrence of falls in older adults. In addition, VO_2max_ may have a direct protective effect against falls through maintenance of the brain structures responsible for balance, especially the frontal cortex and hippocampus, and through its positive influence on cognition [40]. In the present study, the control group had better cognition and executive function, thus reinforcing the central benefit model theory [6,21].

Executive function is one of the main variables involved in falls or the risk of falling. It is hypothesized that impaired executive function can increase the risk of falls via various pathways, including impaired balance and gait, reduced attention capacity, impaired central processing, and impaired execution of postural responses. Muir et al. showed that healthy older adults with poor executive function had a 44% higher risk of falling compared to non-impaired controls [41]. In patients with CVD, this risk increases by an additional 20% when compared to healthy controls [42]. The results of our study showed that the case group had a poorer performance in the trail-making B and verbal fluency tests, thus implicating executive function as a biomarker of the risk of falling, not only when comparing healthy older adults with individuals with CVD, but also in patients receiving treatment. This highlights the importance of management, monitoring, and screening for cognitive decline in this population, especially executive domains, including visuospatial skills and processing speed.

As expected, our study demonstrated age- and medication prescription-related effects on the risk of falling. The participants in the case group were older and took more antihypertensive, beta-blocker, anticoagulant and diuretic medicines than those in the control group. In addition, the random forest analysis showed that not only medication use, but also the combinations of the principal types involved in the prescription (mainly the junction of beta-blockers, anticoagulants, and diuretics) can contribute to the risk of falls in these individuals. Ye et al. (2020) discovered that individuals taking beta-blockers and diuretics are at high-risk of falls, with these medications increasing the risks by 48% and three-fold, respectively. Ye et al. also found that the oldest subjects had a six-fold higher chance of falling over a period of 94 days than younger participants [38]. The AHA Scientific Statement on Preventing and Managing Falls in Adults with Cardiovascular Disease (2022) recommends the use of the lowest possible dose of these medications, and advises on the risk of polypharmacy, especially in older adults, as well as the importance of the management of medicines to mitigate the risk of falling in patients with CVDs [12].

The present study has some limitations and strengths that must be considered. First, the specific characteristics of the sample (mainly white, highly educated and physically active participants) may impair the generalization of our results [1,12]. Second, we did not investigate some key risk factors for falls such as the biological, socioeconomic, environmental, and behavioral dimensions [43]. However, the random forest analysis did not show age, an important biological risk factor, as one of the most important variables associated with the incidence of falls.

## 5. Conclusions

In this study we showed that DT performance, balance, and aerobic capacity levels are associated with an increased risk of the occurrence of falls in older adults with CVDs. Our study also highlights important considerations relating to the role of cardiovascular capacity, balance, and medication in the incidence of falls in older adults with CVDs. Due to the association between cognitive-motor performance and falls identified in this study, future studies should investigate the possible relationship between modifiable risk factors and fall events in older adults with CVDs. Hence, improving DT performance and aerobic physical exercise levels are highlighted as important strategies for preventing falls in older adults with CVDs.

## Figures and Tables

**Figure 1 behavsci-13-00488-f001:**
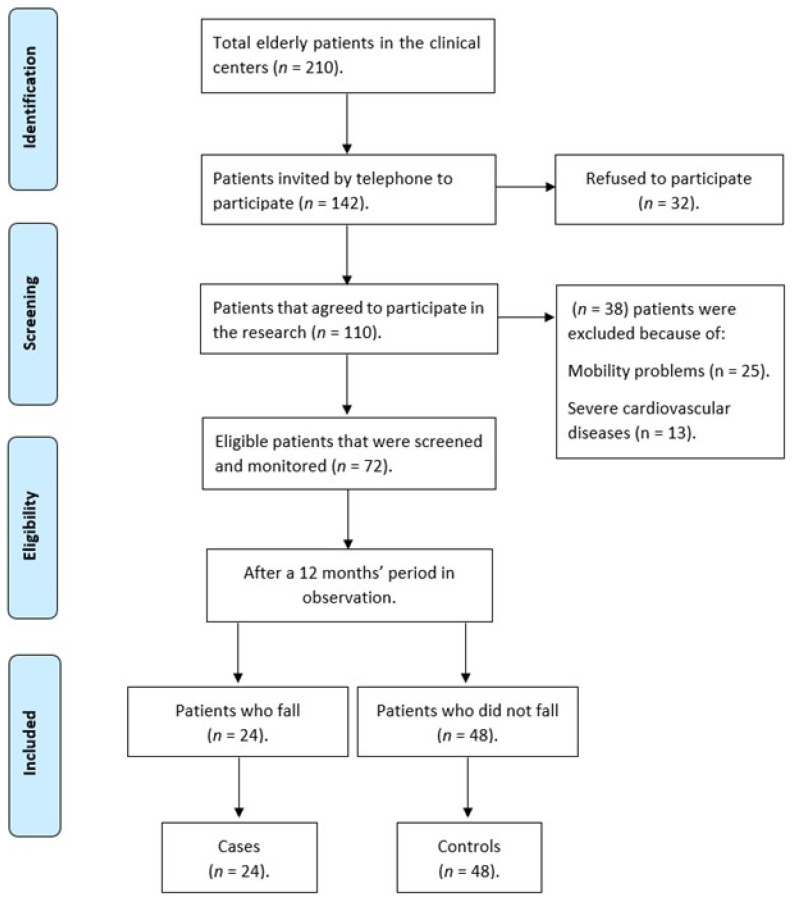
Flowchart of participants included in the case and control groups.

**Figure 2 behavsci-13-00488-f002:**
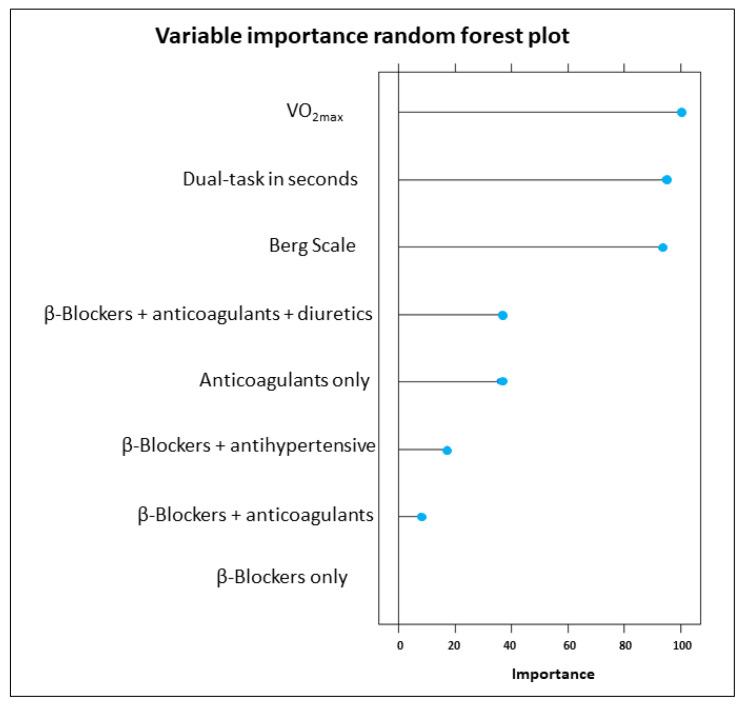
Variable importance based on mean decrease in accuracy.

**Table 1 behavsci-13-00488-t001:** Demographic and clinical characteristics of the groups.

	Case *n* = 24	Control *n* = 48	T/U/*χ*^2^ (*p*-Value)
Age (years) ^b^	76.5 (65.0–92.0)	73.0 (60.0–86.0)	398.5 (0.03) *
Male (%)	15 (62%)	28 (58%)	0.11 (0.73)
Ethnicity			0.85 (0.35)
Black (%)	5 (27%)	6 (14%)	
White (%)	19 (73%)	42 (83%)	
Schooling (years) ^b^	16 (8–22)	16 (8–24)	533.0 (0.59)
Physical activity level			2.89 (0.23)
Inactive (%)	4 (16%)	13 (27%)	
Two times per week (%)	10 (42%)	11 (23%)	
Three times per week (%)	10 (42%)	24 (50%)	
BMI (kg/m^2^) ^b^	26.8 (20.1–45.4)	26.7 (18.1–34.3)	560.5 (0.85)
Disease duration (years) ^b^	10 (2–25)	8 (2–41)	691.0 (0.17)
ACC/AHA classification			11.53 (<0.01) **
B (%)	10 (42%)	39 (81%)	
C (%)	14 (58%)	9 (19%)	
GDS (score) ^b^	2.5 (0–16.0)	1.0 (0–28.0)	455.5 (0.17)
MMSE (score) ^b^	29 (22–30)	29 (23–30)	482.5 (0.30)
Verbal fluency (score) ^a^	18.88 (4.56)	22.25 (5.85)	−2.69 (<0.01) **
Trail-making A (seconds) ^b^	45.10 (29.2–148.9)	40.30 (20.8–86.1)	453.0 (0.17)
Trail-making B (seconds) ^a^	155.44 (86.89)	93.5 (52.0)	3.21 (<0.01) **
Dual-task (seconds) ^b^	8.7 (6.9–20.0)	7.45 (4.9–19.0)	255.5 (<0.01) **
Words DT (*n* animals) ^b^	6.5 (3.0–10)	6 (3.0–9.0)	500.5 (0.96)
TUG (seconds) ^b^	7.44 (5.8–11.0)	6.84 (4.2–11.0)	339.5 (<0.01) **
STS (repetitions) ^b^	9 (5.0–13.0)	11.0 (0–21.0)	364.5 (0.04) *
Handgrip strength (kg) ^b^	23.1 (12.0–52.9)	29.25 (15.5–52.8)	473.0 (0.21)
Berg (score) ^a^	51.71 (4.76)	54.62 (2.63)	−2.79 (<0.01) **
VSAQ (score) ^b^	6.0 (3.0–11.0)	7.0 (4.0–11.0)	436.5 (0.08)
FES-I (score) ^b^	18.5 (16.0–39.0)	16.0 (16.0–32.0)	399.0 (0.03) *
Upper limb flexibility (cm) ^b^	−13.0 (−35.0–0)	−13.0 (−36.0–7.0)	433.5 (0.69)
Lower limb flexibility (cm) ^b^	−12.0 (−33.0–3.0)	−13.0 (−46.0–8.0)	519.5 (0.91)
VO_2max_ (mL/kg/min) ^b^	14.6 (10–23.1)	18.6 (10–29.1)	233.0 (<0.01) **
Cardiac medication use			18.65 (<0.01) **
Antihypertensive	16 (70%)	18 (38%)	
Beta-blockers	24 (100%)	41 (87%)	
Anticoagulants	19 (79%)	12 (25%)	
Diuretics	14 (58%)	9 (20%)	
Psychiatric medication use			8.49 (0.07)
Antidepressants	2 (8%)	3 (6%)	
Anxiolytics	1 (4%)	1 (2%)	
Benzodiazepines	13 (54%)	15 (32%)	
Combined Ant + Benz	4 (17%)	4 (8%)	

Note. BMI: body mass index; ACC/AHA: American College of Cardiology/American Heart Association; GDS: Geriatric Depression Scale; MMSE: Mini-Mental State Exam; DT: Dual-task; TUG: timed up-and-go; STS: sit-to-stand; Berg Balance Scale; VSAQ: Veterans Specific Activity Questionnaire; FES-I: Falls Efficacy Scale International version; VO_2max_: maximum rate of oxygen consumption; Ant + Benz: Combined antidepressants and benzodiazepines. ^a^ parametric variable: mean (standard deviation); ^b^ non-parametric variables: median (minimum–maximum). * *p* < 0.05, ** *p* < 0.01.

**Table 2 behavsci-13-00488-t002:** Association between the occurrence of falls and the most important physical components in the random forest analysis.

Dual-Task			VO_2max_			Berg Scale		
OR	CI_95%_	*p*	OR	CI_95%_	*p*	OR	CI_95%_	*p*
0.85	0.6–1.1	0.3	1.07	0.8–1.3	0.5	1.69	1.2–2.8	0.01 *
0.88 ^#^	0.6–1.2	0.4	1.10 ^#^	0.8–1.3	0.41	1.64 ^#^	1.1–2.4	0.01 *

Note. OR = odds ratio; CI = confidence interval; ^#^ Adjusted for age and lower limb strength (LLS); * *p* ≤ 0.05.

## Data Availability

The data presented in this study are available on request from the corresponding author. The data are not publicly available due to the clinical settings privacy.

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
