# Peer review of "Dual-Task Performance, Balance and Aerobic Capacity as Predictors of Falls in Older Adults with Cardiovascular Disease: A Comparative Study"

_behavsci, 2023, doi:10.3390/bs13060488_

Round 1

Reviewer 1 Report

Thank you for inviting me as a reviewer of this valuable manuscript. This study focused on 'Dual-task performance, balance and aerobic capacity as a pre- 2

dictor of falls in older adults with cardiovascular disease: a prospective case-control by density study'. In order to improve the quality of the paper, I provide the following comments.

(Comment 1) The sample size is too small, making it difficult for correlation analysis results to be meaningful. In general, big-data analysis is performed by setting the experimental group and control conditions with national data. This point should be mentioned in the limitation. Also, I recommend changing the title to:

- Dual-task performance, balance and aerobic capacity as a pre- 2

dictor of falls in older adults with cardiovascular disease: a pilot study

(Comment 2) Due to the different living conditions of the patients included in the study, the results of this study cannot be generalized. Please mention this part in Discussion along with previous research.

Minor editing of English language required

Author Response

We want to highlight that we paid for a native-speaker language review, as suggested. Thank you. 

Reviewer 2 Report

INTRODUCTION

I do not think that fall risk or frailty can be considered as geriatric syndromes. Please, rephrase.

The distinction between modifiable and non modifiable factors are not clearly explained. For instance, cognitive and physical decline can be partially impacted by therapies such as exercise, which help to slow the rate of decline. I mean, its course its somehow modifiable.

Lines 65-66. Implying that CVD is related to falls calls for a definition of falls. Clearly, as  a result of a cardiac event, conscience can be lost and a fall occurs. Nevertheless, these falls are not the same, and do not have the same cause, as those falls related to poor balance, gait impairment, vision loss, etc.

Please, include a definition of dual task.

METHODS

Since the authors are building their case on CVD drugs as a fall risk factor, using drug therapy for CVD should be an inclusion criteria.

How was the incidence of falls during the last 12 months registered?

There seems to be a lot of measures for being performed all together during one hour. These participants really showed a good fitness level, specially taking into account that they had CDV.

Machine learning technique is an important approach in this study. Information in this regard should be shown in the introduction section. Definition of this method and why it is important to use it in studies like the present one.

RESULTS

The explanation of the study design should be presented in the methods section. By the way, it is somehow pretentious to label this study as a nested case control with such a small sample size. Additionally, literature often refers that in nested cases control, for each case, 4-5 controls must be assigned (see PMID: 7845919). Futher, controls should be selected at random, which is not clear if in this study such procedure was followed.  I suggest to label the study as a comparative one.

Patients were excluded due to major CVD…please, clarify. Did their condition worsened during the study?

Figure 1, there is a typo (felt)

How was maximal oxygen consumption obtained? Please, explain.

Table 1 should show pharmacological therapy (as previously commented)

Lines 256. How come there were significant differences between the groups for the prevalence of subjects classified in the “C” category. Did not were the subjects equally matched at the beginning of the investigation according to all relevant characteristics?

Maximal oxygen consumption appears to be the variable that shows the highest relevance. If this variable was estimated by means of the 6MWT, the authors should known that balance, gait impairment and functional autonomy play an important role in the performance of this test. This should be further discussed

I suggest using “case” instead of “faller” group.

DICUSSION

Lines 326-328. There was no information on drug therapy throughout the study that allow for affirming this idea.

This manuscript is more a comparative study than a nested case control study. I provide some suggestions to the authors since the obtained data can be of interest.

Author Response

(The authors gave the same response as above.)

Round 2

Reviewer 2 Report

I do not have further commentaries.

English is fine